# The Changing Biogeography of the Ligurian Sea: Seawater Warming and Further Records of Southern Species

Annalisa Azzola [1,2,*], Carlo Nike Bianchi [3], Lorenzo Merotto [4], Alessandro Nota [5,6], Francesco Tiralongo [6,7,8], Carla Morri [3] and Alice Oprandi [1]

1   Department of Earth, Environment and Life Sciences (DiSTAV), University of Genoa, Corso Europa 26, 16132 Genova, Italy; alice.oprandi@edu.unige.it
2   National Biodiversity Future Center (NBFC), Piazza Marina 61, 90133 Palermo, Italy
3   Department of Integrative Marine Ecology (EMI), Stazione Zoologica Anton Dohrn—National Institute of Marine Biology, Ecology and Biotechnology, Genoa Marine Centre (GMC), Villa del Principe, Piazza del Principe 4, 16126 Genova, Italy; carlonike.bianchi.ge@gmail.com (C.N.B.); carla.morri.ge@gmail.com (C.M.)
4   Marine Protected Area of Portofino, Viale Rainusso 1, 16038 Santa Margherita Ligure, Italy; lorenzomerotto@gmail.com
5   Department of Biology and Biotechnology, University of Pavia, Via Ferrata 9, 27100 Pavia, Italy; alessandro.nota@conted.ox.ac.uk
6   Ente Fauna Marina Mediterranea, Scientific Organization for Research and Conservation of Marine Biodiversity, Via Rapisardi trav. VIII 2, 96012 Avola, Italy; francesco.tiralongo@unict.it
7   Department of Biological, Geological and Environmental Sciences, University of Catania, Via Androne 81, 95124 Catania, Italy
8   Institute for Biological Resources and Marine Biotechnologies, National Research Council, Largo Fiera della Pesca 2, 60125 Ancona, Italy
*   Correspondence: annalisa.azzola@gmail.com; Tel.: +39-010-3538584

**Abstract:** Global warming is causing poleward expansion of species ranges. Temperate seas, in particular, are undergoing a process known as 'tropicalisation', i.e., the combination of sea-water warming and establishment of southern species. The Ligurian Sea is one of the coldest sectors of the Mediterranean and has thus been characterized by a dearth of warm-temperate species and a comparative abundance of cold-temperate species. This paper uses a time series of sea surface temperature (SST) and new records of thermophilic fish species to reconsider the biogeography of the Ligurian Sea. SST has risen by about 0.7 °C on average between 1948 and 2023, but two phases may be distinguished: a cool one (ended in the mid-1980s) and a warm one (still ongoing); the latter phase shows alternating periods of rapid warming and comparatively stationary temperature. The arrival of thermophilic species coincided with the periods of rapid warming; some of these species were established in the subsequent stationary periods. Heatwaves and climate-related diseases associated with the periods of rapid warming have caused mass mortalities of autochthonous species. Our knowledge on the biogeography of the Ligurian Sea was established during the cool phase; the present situation, however, calls for re-defining the chorological spectrum of the Ligurian Sea biota.

**Keywords:** climate change; sea surface temperature; range expansion; *Diplodus cervinus*; *Katsuwonus pelamis*; *Kyphosus vaigiensis*; *Mycteroperca rubra*; tropicalization; Mediterranean Sea





## 1. Introduction

One of the most easily perceived impacts of global warming on biodiversity is the poleward range expansion of both terrestrial and marine species [1,2]. This phenomenon is particularly obvious in the case of marine organisms, whose ranges expand at a faster rate with respect to the terrestrial ones [3–5]. Temperate seas, where the biota is already adapted to the large seasonal variability of temperature at mid-latitudes, provide the best-known examples [6,7]. In particular, temperate seas are experiencing what has been called 'tropicalisation' (British English) or 'tropicalization' (American English), a neologism

coined to indicate temperature increase and the concurrent arrival and establishment of (sub)tropical species, which may lead to changes in the regional marine chorological spectrum [8,9].

Examples of tropicalization of temperate marine regions come from many parts of the world, including North America [10], Australia [11], Europe [12], the Eastern Atlantic [13], Macaronesia [14], and the Mediterranean Sea [15].

The Mediterranean, the largest warm-temperate sea in the world, is a semi-enclosed basin with a small volume-to-surface area ratio, which makes it respond faster and more strongly to warming than the global ocean [16]. It has therefore been defined as a climate change hotspot [17,18], where seawater temperature is increasing at a rate of around $0.04\ °C \cdot a^{-1}$ [19].

Although it is only 0.82% by surface area and 0.32% by volume of the world ocean, the Mediterranean Sea exhibits an astonishing biodiversity, with somewhere between 4% and 18% of the world marine species and a number of endemics averaging more than one quarter of the whole Mediterranean Sea biota [20]. However, this astonishing biodiversity is presently threatened by climate change. Seawater warming favors the establishment of exotic species of (sub)tropical origin [21–23] and drives endemic species to the brink of extinction [24]; frequent marine heat waves (discrete periods of extreme local seawater warming), in particular, are causing mass mortality of native species [25–27]. These threats are even more evident in the Ligurian Sea [28], located at the north-western corner of the Mediterranean. It is therefore comparatively colder and characterized by the presence of some species from cold-temperate waters generally missing elsewhere in the Mediterranean and by the scant occurrence of warm-water species [29]. This gives the Ligurian Sea a moderate boreal affinity [30]. This scenario, however, is changing; in concomitance with seawater warming, the occurrence of warm-water species of various origin in the Ligurian Sea became frequent after the mid-1980s [31–36]. Temperature, however, is not the sole factor facilitating the establishment of thermophilic species; other components of their ecological niche, including salinity and productivity, may also be important [37,38].

The aim of this paper is twofold: (i) first, it analyses the time series of Ligurian Sea temperature to describe its pattern of increase; (ii) second, it reports on the occurrence of four thermophilic fish species hitherto rarely or never found in the Ligurian Sea: the zebra seabream *Diplodus cervinus* (Lowe 1838), the skipjack tuna *Katsuwonus pelamis* (Linnaeus 1758), the brassy chub *Kyphosus vaigiensis* (Quoy & Gaimard 1825), and the mottled grouper *Mycteroperca rubra* (Bloch 1793). The results are discussed in the frame of comparable global changes and integrated within a short review of the current situation of the Ligurian Sea biota, with the prospect of evaluating whether it is currently necessary to reconsider its chorology and biogeographic setting [39].

## 2. Materials and Methods

### 2.1. Ligurian Sea Temperature

Sea surface temperature (SST) records for the period 1948 to 2023 were obtained from the US National Oceanic and Atmospheric Administration (NOAA) satellite data [40]. Downloaded data were calibrated with the discontinuous field measurements (diving computer) available using linear regression (y = 1.0916x + 0.6432; R² = 0.9694) [41]. Sea temperature data before 1948 stored in hydrographical data banks, such as the Mediterranean data bank at the Marine Environment Research Centre of La Spezia (Italy), which contains records since 1909 [42], are unfortunately too inhomogeneous to reconstruct a time series [33].

The overall trend in SST between 1948 and 2023 was illustrated by simple linear interpolation, while smoothing the data by moving averages over seven-year periods was employed to detect major irregularities within the known interannual variability [43].

Warm-water fish species were spotted and photographed by scuba diving or caught by angling or spearfishing between 2016 and 2023, mostly around the Portofino Promontory

or off Genoa, a city overlooking the northernmost stretch of the Ligurian Sea (Figure 1); both localities are situated further north than 44°17.50′ N.

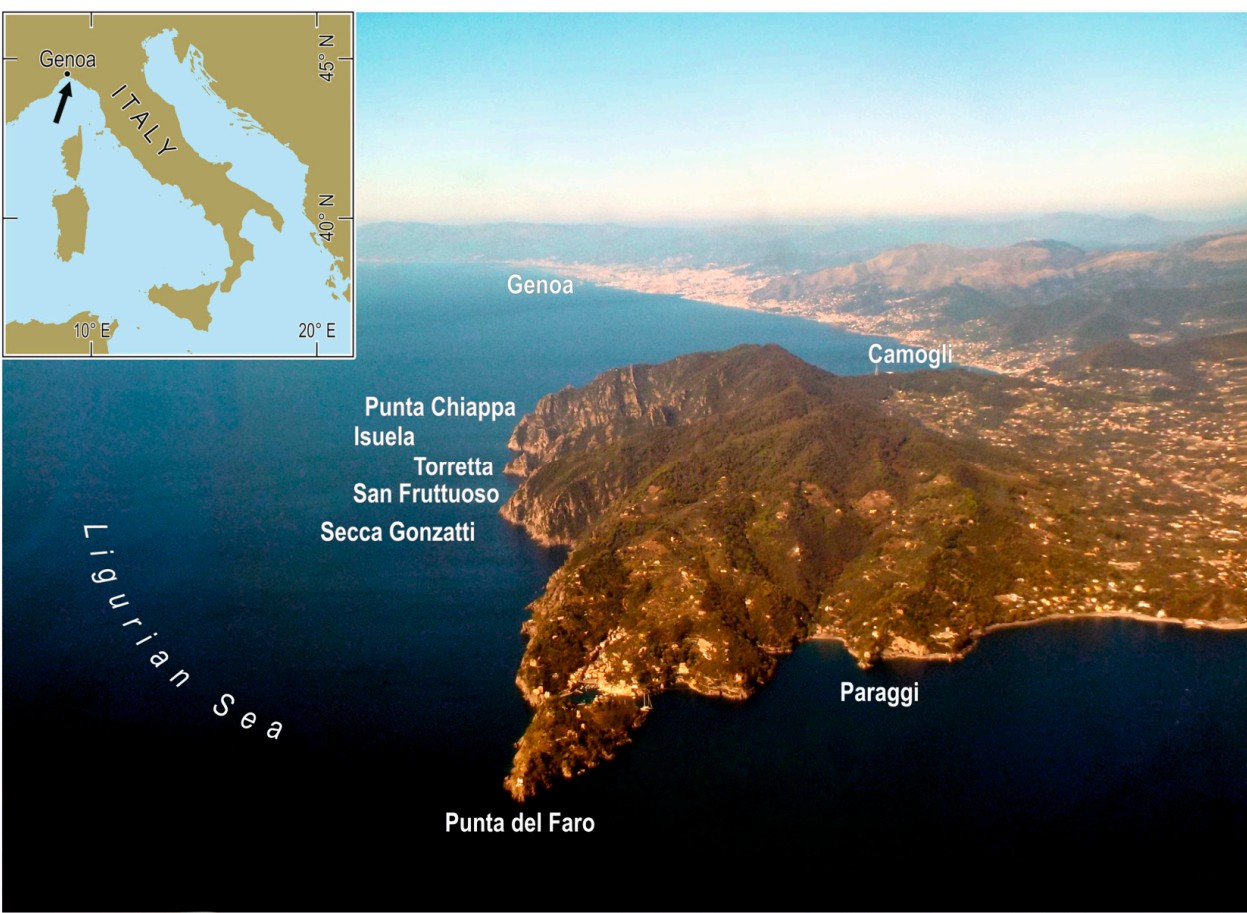

**Figure 1.** Oblique aerial view of the study area, with inset showing its position in Italy. The localities where the warm-water fish were recorded are indicated. The rocky head in the foreground is the Portofino Promontory.

The Portofino Promontory has been a marine protected area (MPA) since 1999 [44], but both fishing and diving are allowed (although with restrictions) in the sites from where the present records of warm-water fish were taken.

*2.2. Fish Species Identification*

*Diplodus cervinus* (Figure 2a), *Katsuwonus pelamis* (Figure 2b), and *Mycteroperca rubra* (Figure 2d) were identified morphologically on the basis of specimens caught or underwater photographs (which also helped confirm additional visual records), according to the relevant volume of the "Fauna of Italy" [45].

On the contrary, the identification of *Kyphosus vaigiensis* (Figure 2c), not included in the above-mentioned volume, required further effort. Two species of *Kyphosus* are known for the Mediterranean Sea [46,47]: the Bermuda chub *Kyphosus sectatrix* (Linnaeus 1758), sometimes erroneously named *Kyphosus saltatrix* (Linnaeus 1758) or *Kyphosus sectator* (Linnaeus 1758), and the brassy chub *Kyphosus vaigiensis* (Quoy & Gaimard 1825), previously reported under the name *Kyphosus incisor* (Cuvier 1831), a junior synonym [48]. On a mere morphological basis, the specimen dealt with in the present paper was initially identified as *K. sectatrix* because of the head profile with a distinct bump in front of and above the eye, not gently convex as it is in *K. vaigiensis* [49]. However, the taxonomy of the genus is problematic [46–51], and delimiting the different species by morphological traits alone may

be deceptive; the application of genetic analyses and molecular taxonomy techniques has therefore been recommended in literature [52,53].

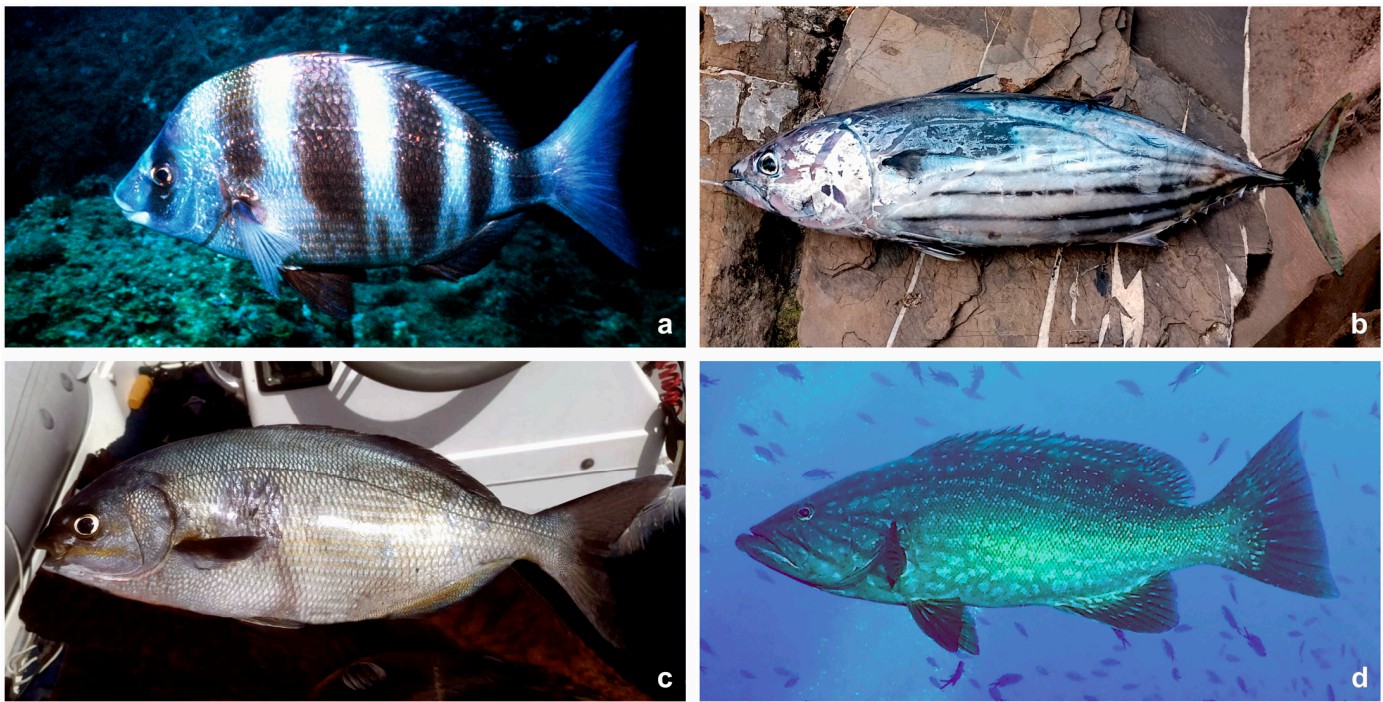

**Figure 2.** Four warm-water fish species recently found in the Ligurian Sea: the zebra seabream *Diplodus cervinus* (Lowe 1838) (**a**); the skipjack tuna *Katsuwonus pelamis* (Linnaeus 1758) (**b**); the brassy chub *Kyphosus vaigiensis* (Quoy & Gaimard 1825) (**c**); and the mottled grouper *Mycteroperca rubra* (Bloch 1793) (**d**).

*2.3. DNA Extraction, Amplification, and Sequencing for Kyphosus vaigiensis*

A fresh tissue sample of the recently collected *Kyphosus* was analyzed genetically in the frame of the Alien-Fish Project [54]. The sample was preserved in absolute alcohol. For the barcoding procedure, it was first rehydrated for 10 min in 1 mL of Jaenisch solution (10 mM Tris-HCl pH 8.5, 30 mM NaCl, and 5 mM EDTA) and then digested overnight in a lysis solution containing 725 µL of Jaenisch solution, 15 µL of proteinase K (at a final concentration of 200 µg·mL$^{-1}$), and 15 µL of SDS 10%. The day after, proteins were removed by the addition of 750 µL of chloroform; after 10 s of vortex followed by 10 min of centrifugation at 14,000 rpm, the supernatant was collected in a new Eppendorf. Total DNA was precipitated by adding an equal volume of isopropylic alcohol followed by centrifugation at 14,000 rpm for 20 min. The pellet was then purified by adding 500 µL of 75% ethanol, dried at room temperature for 1 h, and resuspended in 50 µL of Tris-EDTA buffer. A fragment of mt-COI was amplified with primers based on Folmer et al. [55] and modified as in Astrin and Stüben [56]: fw: LCO1490-JJ, CHACWAAYCATAAAGATATYGG; rev: HCO2198-JJ, AWACTTCVGGRTGVCCAAARAATCA. Polymerase chain reaction (PCR) was conducted in a reaction volume of 30 µL containing 1X reaction buffer, 1.5 mM MgCl2, 5% DMSO, 250 µM dNTPs, 0.5 µM of each primer, and 1 U·sample$^{-1}$ of HotStarTaq Master Mix (Qiagen). DNA was amplified as follows: 15 min of initial denaturation (95 °C) followed by 10 cycles of 30 s at 94 °C, 45 s at 60 °C to 50 °C (lowering the annealing temperature in each cycle by 1 °C), 2 min at 72 °C followed by 30 cycles of 30 s at 94 °C, 45 s at 50 °C, 2 min at 72 °C, and a final extension cycle of 15 min at 72 °C. A total of 5 µL of the amplification product was detected with ethidium bromide on a 3% agarose gel electrophoresis. Both purification and sequencing were performed by an external service (Genechron, Rome). Both strands were sequenced. The amplified COI fragment of the fish

showed 100% identity with previously published sequences belonging to the species *K. vaigiensis* [46,50].

### 2.4. Fish Species Ranges

The guides to the fishes of the Mediterranean and Black Seas [57] and of the Northeastern Atlantic and the Mediterranean [58] and the atlas of exotic fishes in the Mediterranean Sea [59], integrated with specific publications [47,50,54,60–78], have been used to draw species ranges and/or disjunct occurrences within the Mediterranean Sea. Records of *K. saltatrix* in Italy [79] and Libya [80], later recognized to concern most probably *K. vaigiensis* [46,61], were included among those of the latter.

## 3. Results

### 3.1. Temperature

NOAA satellite yearly means of SST between 1948 and 2023 clearly exhibited a warming trend: linear fitting would indicate an average increase of about 0.7 °C in the last 75 years (Figure 3). However, the trend is not linear, but it shows at least two major phases, notwithstanding high year-to-year variability: (i) a cooling phase roughly between 1964 and 1984, when yearly average temperatures dropped from ca 18.6 °C to ca 17.6 °C, and (ii) a warming phase since 1985 to reach the present yearly average of over 19.3 °C. In turn, alternating periods of rapid warming and stationary temperatures were recognizable within the warming phase (Figure 3). In the first period of rapid warming, between 1985 and 1992, SST rose by 0.08 °C·a$^{-1}$ and remained comparatively stable around 18.2 °C until 1998. A second period of rapid warming occurred between 1999 and 2006, with the SST increasing by 0.07 °C·a$^{-1}$ on average; then, SST exhibited little variation around 18.7 °C until 2013. The third period of rapid warming started in 2014 and is still going on, with a rate of over 0.06 °C·a$^{-1}$.

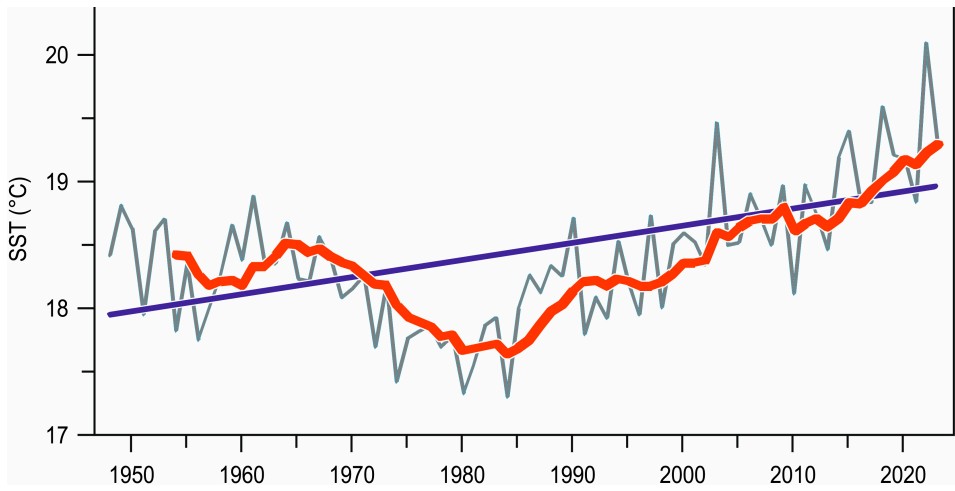

**Figure 3.** Trend of NOAA satellite data of sea surface temperature (SST) for the Ligurian Sea between 1948 and 2023. The thin broken line illustrates the annual means, the straight line represents the linear interpolation, and the thick, broken line depicts the moving average smoothed over seven years.

### 3.2. Warm-Water Fishes

An individual of *Diplodus cervinus* (Figures 2a and 4a) of ca 25 cm was spotted on 21 June 2020 at a 15 m depth in Portofino MPA in the diving site named Torretta. In the same year, a larger specimen, with a total length of 32 cm, was fished off Camogli (Supplementary Materials Figure S1) and another off the breakwater of the Port of Genoa. Two further specimens were sighted by divers in localities east of Portofino in 2021 and 2022. A sixth record was an individual of ca 15 cm seen at 5 m at Paraggi in June 2023.

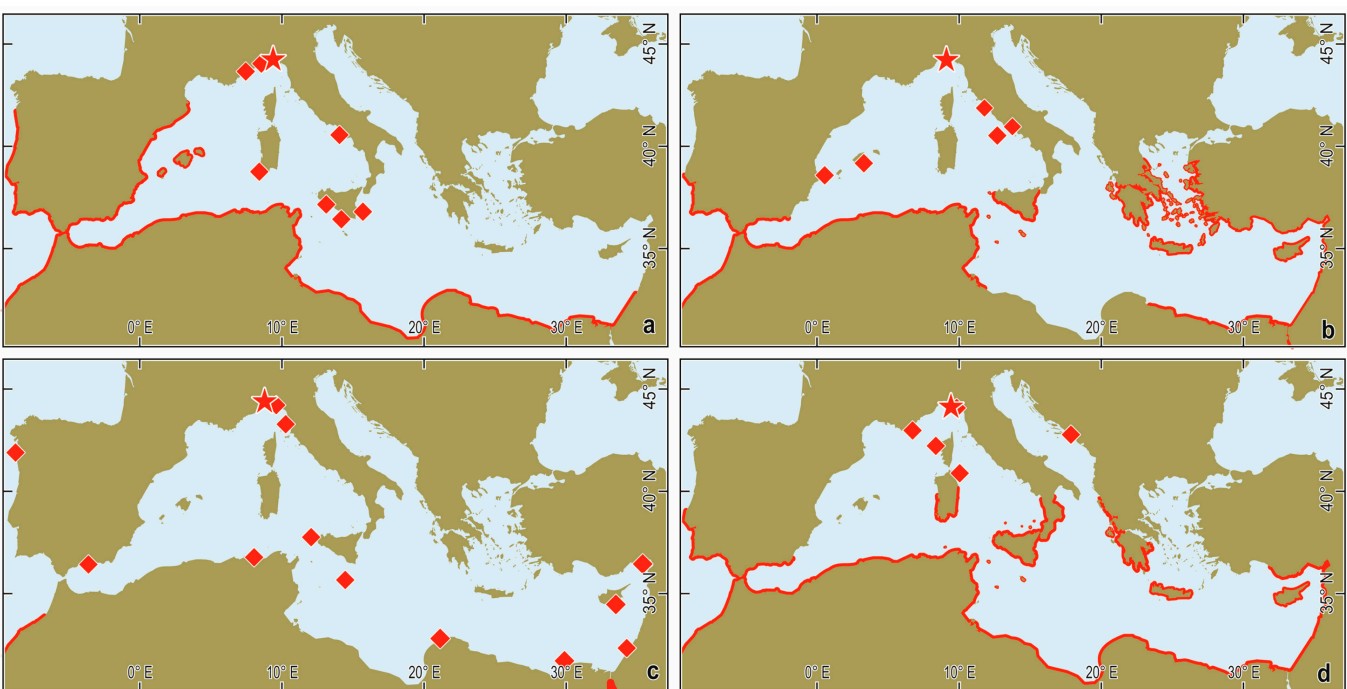

**Figure 4.** Mediterranean occurrences of *Diplodus cervinus* (**a**), *Katsuwonus pelamis* (**b**), *Kyphosus vaigiensis* (**c**), and *Mycteroperca rubra* (**d**). The thick, continuous line depicts the historical range; diamonds are disjunct records (see text for references); and stars indicate the present records.

A small school of three individuals of *Katsuwonus pelamis* (Figures 2b and 4b) about 40 cm long was observed on 17 October 2020 at a 5 m depth in San Fruttuoso Creek. A specimen of 50 cm, weighing 1.7 kg, was fished at Punta Chiappa on 14 November 2023 amid several specimens of the mackerel *Euthynnus alletteratus* (Rafinesque 1810).

A single specimen of *Kyphosus vaigiensis* (Figures 2c and 4c), of 2 kg weight, 52 cm total length, and 47.5 cm fork length, was spearfished on 13 September 2022 at a 7 m depth off the breakwater of the Port of Genoa (Supplementary Materials Figure S2).

*Mycteroperca rubra* (Figures 2d and 4d) has been frequently observed in the area of Portofino in the last years (Supplementary Materials Figures S3–S5). It was first sighted by scuba divers in 2016 and then regularly seen starting from 2020 to date (Table 1). Most records concerned adults of about 40 cm in length. It was most commonly encountered in exposed sites, such as rocky shoals (Secca Gonzatti, Isuela) and rocky points (Punta del Faro), where it appeared fully integrated into the native fish fauna, such as the white sea-bream *Diplodus sargus* (Linnaeus 1758), the two-banded sea bream *Diplodus vulgaris* (E. Geoffroy Saint-Hilaire 1817), the black sea bream *Spondyliosoma cantharus* (Linnaeus 1758), the painted comber *Serranus scriba* (Linnaeus 1758), the brown meagre *Sciaena umbra* (Linnaeus 1758), and the damselfish *Chromis chromis* (Linnaeus 1758), among others (Figure 5). In the Portofino MPA, *M. rubra* coexists with the native and by far more abundant dusky grouper *Epinephelus marginatus* (Lowe 1834), which, however, is more commonly observed below 15 m.

**Table 1.** Sightings of *Mycteroperca rubra* by scuba divers in Portofino MPA.

| Date | Site | Depth (m) | Number of Individuals |
|---|---|---|---|
| 3 August 2016 | Secca Gonzatti | 15 | 1 |
| 29 July 2020 | Secca Gonzatti | 8 | 1 |
| 6 June 2020 | Isuela | 15 | 1 |
| 9 September 2020 | Isuela | 20 | 1 |
| 12 June 2021 | Secca Gonzatti | 5 | 1 |
| 15 June 2021 | Punta del Faro | 10 | 1 |
| 19 June 2021 | Secca Gonzatti | 5 | 5 |
| 24 June 2021 | Isuela | 15 | 1 |
| 11 July 2021 | Secca Gonzatti | 5 | 1 |
| 21 July 2021 | Secca Gonzatti | 10 | 2 |
| 17 June 2023 | Secca Gonzatti | 7 | 1 |
| 30 July 2023 | Isuela | 15 | 1 |
| 17 September 2023 | Punta del Faro | 5 | 1 |

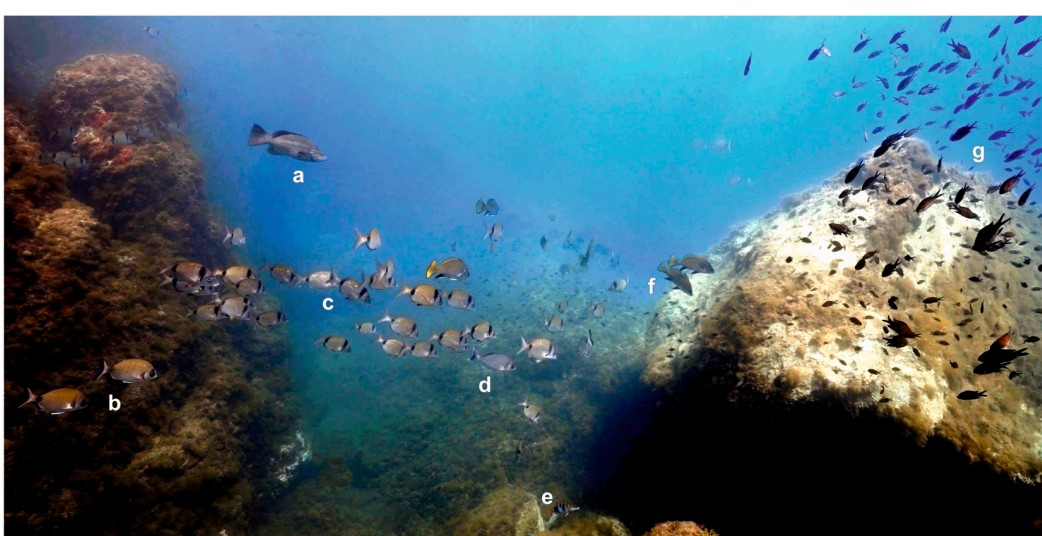

**Figure 5.** *Mycteroperca rubra* (a) swimming amid native fish species in Portofino MPA waters: *Diplodus sargus* (b), *D. vulgaris* (c), *Spondyliosoma cantharus* (d), *Serranus scriba* (e), *Sciaena umbra* (f), and *Chromis chromis* (g).

## 4. Discussion

A cooling phase between 1958 and 1980 within the warming trend of Ligurian Sea temperatures had already been detected in a previous study that analyzed data collected at sea fortnightly off Villefranche-sur-Mer in the French part of the Ligurian Sea [81]. Apparently, this cooling phase also occurred globally; all global temperature series highlight a cooler interval in the 1960s–1970s [82]. This temporary reversal of the general trend has been attributed to the action of sulfur aerosols, emitted by volcanic eruptions but produced in abundance in those years by the combustion of fossil fuels, burnable waste in automobiles, and thermal power plants; these aerosols cooled the Earth's climate via an increase in the Earth's albedo [83]. On the other hand, atmospheric chemistry generated sulfuric acid from these aerosols; eventually, they precipitated as acid rain, causing severe impacts on vegetation, soil, buildings, monuments, and water bodies, as well as on human health [84]. Growing public concern in the 1970s led to coordinated policy actions that substantially reduced sulfur emissions to decrease these impacts [85].

Once acid rain ceased, since the 1980s, average planetary temperatures have started to rise again [86], and this rise is mirrored in the SST series of the Ligurian Sea. However, a pattern not evidenced yet in global data series—but clearly seen in Ligurian Sea SST—

is that temperature is not increasing linearly but with an alternation of rapid rises and comparatively stationary intervals. In global data series, the most recent stationary interval (ca. 2006–2013) was hastily interpreted as a sign that global warming had ended [87], but measurements in the following years showed that temperatures were starting to rise again [88].

Undoubtedly, changing temperature is just one among the many factors driving species occurrence patterns [89]. Nevertheless, records of warm-water species in the Ligurian Sea increased during the rapid warming periods [36,81], while in the subsequent stationary periods, these species have been either able or unable to persist. Of course, it is not the temperature, per se, that facilitates the arrival of warm-water species [90,91], but higher temperatures are instrumental to their establishment success [36,92,93] and play a major role in modulating the magnitude of their ecological impact [94].

In the first period of rapid warming (1985 to 1992), the rainbow wrasse *Thalassoma pavo* (Linnaeus 1758) started to be recorded regularly in the Ligurian Sea [35]; although not as abundant as it is in the southern sectors of the Mediterranean Sea [95], it is now established and reproduces successfully [96–98]. In the second period of rapid warming (1999 to 2006), *Diplodus cervinus* came to the Ligurian Sea and was subsequently established in localities of the western coast [36,98]; the present paper reports its first findings in eastern coast sites, a clue to its further range expansion. Before these recent records, only a specimen of *D. cervinus* was fished in 1896 in the Ligurian Sea [99,100]; roughly in the same years, at the turn of the 19th and 20th centuries, *T. pavo* was also occasionally recorded [96,100], which led to the supposition of a warm climatic phase in those years [21]. No reliable SST data, however, are available to corroborate this hypothesis [33]. The third and most recent period of rapid warming (2014 to present) saw the arrival of *Mycteroperca rubra* in the Ligurian Sea. Scuba divers reported on the recent occurrence of two other warm-water groupers, namely the white grouper *Epinephelus aeneus* (Geoffroy Saint-Hilaire 1817) [101] and the dogtooth grouper *Epinephelus caninus* (Valenciennes 1843) [102]. Not only fish but also warm-water algae and invertebrates, including exotic species, were seen to conform to such temporal pattern [27,31,32,36,93]. Even the only alien seagrass to have penetrated the Mediterranean Sea, *Halophila stipulacea* (Forsskål) (Ascherson 1867), showed a similar trend [103–105]: first recorded at Rhodes, SE Aegean Sea [106], it spread only throughout the eastern Mediterranean until the 1980s–1990s (first rapid warming period) to eventually enter the Tyrrhenian Sea in the early 2000s (second rapid warming period) [107]; between 2018 and 2022 (third rapid warming period), it reached NE Sardinia [108], W Corsica [109], and Cannes on the French part of the Ligurian Sea [110]. Unavoidably, information mostly concerns conspicuous species of direct economic and/or ecological interest, often overlooking smaller motile invertebrates [111] that spread similarly [93].

The four warm-water fish species dealt with in the present paper tell different stories. *Katsuwonus pelamis* is a vagrant circumtropical species, occasionally recorded in the Italian seas [70,112]; like other coastal pelagic species, it is capable of undergoing rapid yet variable poleward range shifts [113]. This fish, important from an economical point of view, occurs above the 15 °C winter isotherm worldwide and has been found as far north as 55° in the eastern Atlantic and as far south as 45° in the western Indian Ocean [114]. Similar considerations can be noted about another coastal pelagic warm-water fish, the blue runner *Caranx crysos* (Mitchill 1815), recently recorded in the Ligurian Sea [115,116]. This carangid is another relevant fishery resource widely distributed across the Atlantic Ocean, from Brazil to Canada in the western part and from Angola to Great Britain in the eastern part, Mediterranean Sea included [117].

*Kyphosus vaigiensis* was not known in the Mediterranean before 1998 [58,67] but is now spreading rapidly throughout the whole basin [50–62,68,69,74,76]. It has been considered an exotic [59] or possibly cryptogenic species [78], since both autonomous spread and human-mediated introduction are possible, members of the family Kyphosidae being known to actively follow vessels [118]; research is pending to clarify its status in the region. The species was initially thought to be restricted to the Indo-Pacific region but was later

recognized to be of circumtropical distribution [48]. The present single record off Genoa was preceded by the record of two specimens off Camogli in 2009 [64], but it cannot be said at present whether the species is getting established in the Ligurian Sea. *K. vaigiensis* adds to the already rich list of non-native fish species that have recently colonized the Ligurian Sea [119,120]; such a list also includes the blue-spotted cornetfish *Fistularia commersonii* (Rüppell, 1838), a Red Sea migrant that possibly reproduces in the Ligurian Sea [36]. After the end of the rapid warming period that promotes their arrival, exotic species may either persist indefinitely, even if with highly variable numbers (natural fluctuation model), or abruptly reduce to virtually disappear (boom and bust model) [121]; the latter possibility might be chiefly expectable if the species does not reproduce sexually in the newly colonized area [122]. Should *K. vaigiensis* become established, the addition of another herbivorous fish species could severely impact the already impoverished macroalgal vegetation of the Ligurian Sea [123–125].

The main candidates to become stable components of the Ligurian Sea fish fauna are the two southerners, *Diplodus cervinus* and *Mycteroperca rubra.* The former is regularly seen in sites of the western Ligurian Sea and is apparently further expanding its range north-eastward; the latter seems already fully established [98]. *D. cervinus* is distributed in the eastern Atlantic coast, from the Bay of Biscay to the Cape Verde Islands, Madeira, and Canary Islands and from Angola to South Africa; it is also present in the warmer areas of the Mediterranean Sea [126]. *Mycteroperca rubra* is distributed along the eastern coast of the Atlantic Ocean, from Portugal to Angola and in the southern Mediterranean Sea. However, its abundance is not uniform in the space within its distribution range (i.e., it is recorded as common in Senegal but considered rare along the North Africa coast [127].

The ongoing modifications of the marine biota will not halt if Ligurian Sea temperatures keep on rising. Seawater warming is a global issue that cannot be addressed at a local scale but requires international actions. Agreements on climate change date back to the Kyoto Protocol in 1997, yet little substantial emission abatement has taken place [128,129]. The Paris Agreement in 2015 has been widely hailed as a breakthrough in global climate cooperation, but its goal of keeping global warming well below 2 °C, above preindustrial levels, is at risk of failure [130,131]. Little public support [132] and opposing economic interests [133] work against climate policies. Most economic evaluations only consider the global emission abatement cost but ignore the potential benefits of avoiding the climate damage [134]. The climate change mitigation policies implemented by some of the major world economies have proven successful [135], but actions that are not supported by all countries cannot be very effective [136]. It is imperative to adopt globally sustainable energy policies, which involve a substantial increase in the use of renewable energy sources coupled with the implementation of eco-friendly industrial practices. Investing in research and innovation for low-carbon technologies will play a crucial role in mitigating global warming [137] and consequent species range shifts.

According to the reports and predictive models of the IPCC (Intergovernmental Panel on Climate Change), even with a fourfold reduction in carbon dioxide emissions from current levels, the temperature would continue to rise, with irreversible consequences for ecosystems [138]. The climate change and biodiversity crises are typically addressed independently, but they are fundamentally connected [139]. Mediterranean native communities do not seem capable of keeping up with the ongoing pace of warming [140]. Genetic adaptive responses of marine species would probably be slower than the rate at which sea temperatures are currently rising [141,142]. Rather, human activities, including fisheries [143], should adapt to warming and the consequent spread of warm-water species.

Local anthropogenic pressure may exacerbate the effects of rising ocean temperatures [144] and non-native species arrival [145]. Regional management practices may help reduce local threats, thus indirectly making ecosystems more resilient to global change [144].

Marine protected areas (MPAs) are universally considered the most important tool to manage and conserve marine ecosystems [146]. According to the International Union for Conservation of Nature (IUCN), MPAs are "clearly defined geographical spaces in the sea,

recognized, dedicated, and managed, through legal or other effective means, to achieve the long-term conservation of nature, with associated ecosystem services and cultural values, and to protect habitats and species from anthropogenic threats, allowing for the sustainable use of marine resources within their boundaries" [147]. MPAs are typically designed to manage habitat use and fisheries [148], not to address all the pressures placed on marine habitats [149]; this notwithstanding, they have proven to be an effective tool to restore ecosystems [150,151] and to enhance their resilience to climate impacts [152] and other disturbances [153]. However, the examples in the Ligurian Sea illustrate that species distribution is changing as a result of climate change, potentially compromising the efficacy of MPAs as biodiversity conservation tools [154,155]; consequently, many studies worldwide suggest that MPAs alone cannot buffer the consequences of ocean warming [156–160]. MPAs are traditionally created under the assumption that the biodiversity they protect is static, which is not the case under a changing climate; as a matter of fact, climate change is inadequately considered in MPA management plans [161]. Precautionary anticipation of the future impacts of climate change on marine biodiversity should inform MPA zoning and regulation [162]. Context-specific management measures should consider pressures that may be both endogenic (caused within the MPA) and exogenic (with causes from outside the MPA, such as climate change and non-native species) [163].

The Ligurian Sea hosts the International Whale Sanctuary, created to protect the pelagic environment, and a number of coastal MPAs established by France, Monaco, and Italy [119]. However, there is little or no coordination among them, and many are small and/or not adequately enforced [164,165]. Large MPAs and well-coordinated MPA networks may allow the incorporation of spatial refugia against climate impacts and offer insurance against local losses [159,161,166,167]. Efficient communication and public participation, two important side products of MPA management [168], would help grow ocean literacy and foster environmental awareness within the local community and among tourists.

## 5. Conclusions

This study illustrated the pattern of seawater warming in the Ligurian Sea, providing suggestive evidence that such a local pattern mirrors the global one. Linking global to regional trends is a basic step for understanding the ecological impacts of marine climate change [169]. In particular, two climatic phases were distinguished: a cool one, ending in the mid-1980s, and a warm one that is still ongoing.

The biota of the Ligurian Sea has been the object of studies since the second half of the XVIII century [170–173], but the bulk of the research that led to outlining its biogeography was carried out in the cool phase [30,174,175]; the main distinguishing characteristics of the Ligurian Sea were said to be the dearth of thermophilic species and the comparative abundance of cold-temperate boreal species [29].

In the warm phase, which roughly started in 1985, an ever-growing number of thermophilic species (either exotic or native to the southern sectors of the Mediterranean) arrived (and are still arriving) in the Ligurian Sea, specifically during the periods of rapid warming. Some of them were established in the subsequent periods of comparatively stationary temperatures. Establishment of warm-water species is abruptly wiping out the first of the two main distinguishing characteristics of the Ligurian Sea biota.

If southern, warm-water species (of whatever origin) settle in the Ligurian Sea, what happens to the native cool-water species thriving there? Are they at risk of extinction [21]? Proving that a species no longer exists in a given area is difficult, especially in the marine environment [176]. Frequent and intense heatwaves associated with the periods of rapid warming have caused mass mortalities of many Ligurian Sea autochthonous species [26,177,178], some of which, however, survive in deeper waters [28,179,180]. Primary productivity alteration [181], together with other dysfunctions [182] and disturbances [183], have heavily impacted the Ligurian Sea biota [28]. Climate-related microbial diseases [184], in particular, have possibly led the iconic fan mussel *Pinna nobilis* (Linnaeus 1758) to extinction [185,186]; the warm-water congeneric *Pinna rudis* (Linnaeus 1758),

hitherto never recorded in the Ligurian Sea, is apparently taking its place [187]. There is evidence that southerners can drive native species to extinction [188]. In the western part of the Ligurian Sea, warm-water crustacean species have replaced their cold-water counterparts [189,190], while in the Portofino MPA, alien species have depressed β-diversity through biotic homogenization [191]. Mortalities, rarefactions, replacements, and reduced diversity are possibly also wiping out the second main distinguishing characteristic of the Ligurian Sea biota.

Thus, the modifications undergone in recent decades are making the Ligurian Sea lose its biogeographic peculiarities and acquire a different configuration, partly shaped by the unprecedented abundance and ubiquity of exotic species [21,36]. While the species range shifts and biological invasions driven by ocean warming and their ecological impacts on recipient ecosystems have been amply documented [192–194], the consequences on the world marine biogeographical regionalization have been little explored. This review represents an example on a regional sea that might be a model for a global phenomenon. It is time to reconsider the chorological spectrum of the Ligurian Sea biota. Besides continuing the surveillance of thermophilic invasive species, future research should assess, in particular, the status of the populations of endemic and boreal species in the once-cool Ligurian Sea.

**Supplementary Materials:** The following supporting information can be downloaded at: https://www.mdpi.com/article/10.3390/d16030159/s1, Figure S1: Diplodus cervinus, of 32 cm total length, caught off Camogli in July 2020 (© Sara Giovanna Guaraglia); Figure S2: Kyphosus vaigiensis, of 2 kg weight, 52 cm total length, and 47.5 cm fork length, spearfished in September 2022 at a 7 m depth off the breakwater of the Port of Genoa (© Stefano Castronovo); Figure S3: Mycteroperca rubra resting on the top of the Isuela Shoal, Marine Protected Area of Portofino, in June 2020 (© Lorenzo Merotto); Figure S4: Mycteroperca rubra in mid-water at Secca Gonzatti, Marine Protected Area of Portofino, in July 2021 (© Giuseppe Galletta); Figure S5: Close-up of Mycteroperca rubra at the Isuela Shoal, Marine Protected Area of Portofino, in June 2021 (© Giuseppe Galletta).

**Author Contributions:** Conceptualization, A.A. and C.N.B.; methodology, C.N.B. and C.M.; software, C.N.B., A.N. and F.T.; validation, L.M., C.M., A.O. and F.T.; formal analysis, C.N.B., C.M., A.N. and F.T.; investigation, A.A., C.N.B., L.M., A.N., A.O. and F.T.; resources, A.A., L.M., A.N., A.O. and F.T.; data curation, A.A., C.N.B., A.N., A.O. and F.T.; writing—original draft preparation, A.A., C.N.B., C.M. and A.N.; writing—review and editing, A.A., C.N.B., L.M., C.M., A.N., A.O. and F.T.; visualization, A.A., C.N.B. and C.M.; supervision, A.A.; project administration, A.A.; funding acquisition, A.A. and C.M. All authors have read and agreed to the published version of the manuscript.

**Funding:** This work was partially funded by the National Recovery and Resilience Plan (NRRP), Mission 4 Component 2 Investment 1.4—Call for tender No. 3138 of 16 December 2021, rectified by Decree No. 3175 of 18 December 2021 of the Italian Ministry of University and Research funded by the European Union—NextGenerationEU. Project code CN_00000033, Spoke 1, Concession Decree No. 1034 of 17 June 2022 adopted by the Italian Ministry of University and Research, Project title "National Biodiversity Future Center—NBFC" (G. Bavestrello).

**Institutional Review Board Statement:** Not applicable.

**Data Availability Statement:** Sea surface temperature data are freely available at www.esrl.noaa.gov/psd/cgi-bin/data/timeseries/timeseries1.pl (accessed on 5 January 2024). Fish record data are included in the paper.

**Acknowledgments:** Sara Giovanna Guaraglia, Sara Liverani, Pietro Tassara, and Eleonora Zanon communicated the records of *Diplodus cervinus*. The specimens of *Katsuwonus pelamis* and *Kyphosus vaigiensis* were caught by Massimo Fasce and Stefano Castronovo, respectively. The late Giuseppe Galletta—an unforgotten friend—provided many underwater photographs of *Mycteroperca rubra*, including the one used in Figure 2d. The image in Figure 5 was extracted from an underwater video shot by Fabio Benelli. The staff of Sub Tribe (Genoa) provided field assistance, and all members of GDA (Genoa) participated in many of the dives. Special thanks are due to Eng Sengsavang, reference archivist of the UNESCO Archives, for their help with the bibliography. C.N.B. and C.M. wish to dedicate this paper to their master of marine biogeography Enrico Tortonese (1911–1987). The study of warm-water species in the Ligurian Sea fell under the frame of the project 'The impacts of biological

invasions and climate change on the biodiversity of the Mediterranean Sea' (Italy–Israel cooperation on environment, research, and development).

**Conflicts of Interest:** The authors declare no conflicts of interest.

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
