# Peer review of "The Changing Biogeography of the Ligurian Sea: Seawater Warming and Further Records of Southern Species"

_diversity, doi:10.3390/d16030159_

Round 1
Reviewer 1 Report
Comments and Suggestions for Authors
The manuscript is well-structured and provides valuable insights into the impact of global warming on the Ligurian Sea, particularly regarding the poleward expansion of species ranges and the phenomenon of tropicalization. However, I would recommend the authors to rearrange it as a review since they actually analyse the existing literature within the investigated area. Data regarding the SST as well as the analysis of this data can be incorporated within this review. If the authors still want to keep it as research article, in the Introduction paragraph authors should define the gaps of their analysis, precisely they are aware that temperature is not only parameter that led to these changes in biodiversity. In Materials you should clarify how the genetic analysis was conducted for Kyphosus vaigiensis. In Disscussion you should focus on examined four species – their origins, the way they reached the Ligurian Sea, what this means for native species, and suggest possible measures of protection.
In general, I find this manuscript worth publishing especially if they are willing to make it as review.
Author Response
Please find the reply in the attached file

Reviewer 2 Report
Comments and Suggestions for Authors
This paper is a fine contribution that describes the ‘tropicalisation¨ of he Ligurian Sea, which is one of the coldest sectors of the Mediterranean.
Despite is crystal clear that the Global warming is causing the sea-water warming and establishment of southern species, there is no anywhere in the paper a plea, a call or even recommendations to be implemented to slow down the the Global warming.
Also, punctual recommendations to protect the native biota of the Ligurian sea, must be addressed.
Comments on the Quality of English Language
This paper is a fine contribution that describes the ‘tropicalisation¨ of he Ligurian Sea, which is one of the coldest sectors of the Mediterranean.
Despite is crystal clear that the Global warming is causing the sea-water warming and establishment of southern species, there is no anywhere in the paper a plea, a call or even recommendations to be implemented to slow down the the Global warming.
Also, punctual recommendations to protect the native biota of the Ligurian sea, must be addressed.
Author Response

(The authors gave the same response as above.)
